# Friction Reduction Achieved by Ultraviolet Illumination on TiO_2_ Surface

**DOI:** 10.3390/ma17071680

**Published:** 2024-04-06

**Authors:** Xiao Sang, Ke Han, Manfu Zhu, Liran Ma

**Affiliations:** State Key Laboratory of Tribology in Advanced Equipment, Tsinghua University, Beijing 100084, China; sangx19@mails.tsinghua.edu.cn (X.S.); hank18@tsinghua.org.cn (K.H.); zmf19@mails.tsinghua.edu.cn (M.Z.)

**Keywords:** friction reduction, oil lubrication, photosensitive materials, TiO_2_

## Abstract

Controlling friction by light field is a low-cost, low-energy, non-polluting method. By applying ultraviolet light on the surface of photosensitive materials, the properties of the friction pairs or lubricant can be influenced, thus achieving the purpose of reducing friction. In this study, TiO_2_, an inorganic photosensitive material, was selected to investigate the modulating effect of light fields on friction lubrication when using polyalphaolefin (PAO) base oil as a lubricant, and the modulation law of light fields on the friction lubrication behavior was investigated under different loads (1–8 N), different speeds (20–380 mm/s), and different viscosities (10.1–108.6 mPa·s) of PAO base oil. The experimental results showed that light treatment could reduce the friction coefficient of PAO4 base oil lubrication from 0.034 to 0.016, with a reduction of 52.9% under conditions of 3 N-load and 56.5 mm/s-speed, and the best regulation effect could be achieved under the mixed lubrication condition. After TiO_2_ was treated with ultraviolet light, due to its photocatalytic property, PAO molecules were oxidized and adsorbed on the TiO_2_ surface to form an adsorption layer, which avoided the direct contact of rough peaks and thus reduced the friction coefficient. This study combines photosensitivity, photocatalysis, and friction, presenting a method to reduce the friction coefficient by applying a light field without changing the friction pairs or lubricants, which provides a new direction for friction modulation and gives new ideas for practical applications.

## 1. Introduction

Friction is very common in life, but it is usually a phenomenon that needs to be avoided in industries and other areas due to a large amount of energy consumption and component failure. In the traditional lubrication system, the reduction of friction can usually be achieved by adjusting the normal load and changing the lubricant or the friction pairs, which means it is usually difficult to achieve real-time and efficient regulation of the system friction. However, by applying an external field to change the physical and chemical properties of the friction pair or lubricant, the lubrication process can be actively regulated. A series of techniques to regulate friction and lubrication using external stimuli such as electric field [1,2,3], magnetic field [4,5,6], light field [7,8,9], temperature [10], and pH [11] have attracted a lot of attention from academia and the industry. Among them, the regulation of friction lubrication by light field has become a research hotspot for active regulation because of its advantages of a low cost, low energy consumption, no pollution, and the fact that it can be applied to environments such as outer space, micromechanics, and biological organisms, which fits the future development of green, intelligent, and efficient demand.

The basis of photoinduced friction lubrication lies in materials with photosensitive properties. The physical and chemical properties of photosensitive materials can be regulated by the light field, which in turn affects the state of the friction pairs or lubricant, thus achieving the purpose of friction regulation. Diarylethene [12,13], azobenzene [14,15,16,17], and spiropyran [18,19] are widely used organic photosensitive materials for their changeable molecular structure, which could influence their tribological performance. Titanium dioxide (TiO_2_) and other inorganic photosensitive materials have been studied and applied as photocatalysts for the oxidative degradation of hydrocarbon organic compounds [20,21,22,23], which have great potential and value for application in the photocatalysis and photodegradation of organic materials due to the redox effect of electron–hole pairs [24,25,26,27]. Researchers have found that adding TiO_2_ particles as an additive to lubricating oils effectively enhances their lubricating properties, with some reducing the friction coefficient by 28.4% [28,29]. The composite material forms a good shear lubrication layer on the surface of the friction pair, thus reducing the friction and wear of the system. Some researchers prepared lubricants with both friction-reducing effects and photocatalytic abilities by adding TiO_2_ particles to them, and the lubricating property and photocatalysis were discussed separately [30]. There have also been studies combining light treatment with friction, and it has been found that the friction coefficient of TiO_2_ surfaces can be altered by light exposure through the mechanism of atomic interactions between the friction pairs or the transformation of the structure of the surface water molecules [31,32,33,34]. The above studies mainly focus on the photocatalytic properties of titanium dioxide and its lubricating properties as additives. It has been found that titanium dioxide can be affected by light to change its tribological properties, but few researchers have combined its photocatalytic and lubricating properties together and there is not a systematic study on the effect of light on the friction properties of the theoretical model.

In our previous research, the mechanism of UV light on TiO_2_ in reducing friction when using water and propanetriol as lubricants and the important role that photocatalysis plays in photo-induced friction reduction has been investigated [35]. However, in practical working conditions, the low viscosity of water makes film formation difficult and oil lubrication is more common, so it is more meaningful to study oil lubrication. In this research, the effect of a light field on TiO_2_ in different systems where polyalphaolefin (PAO) base oil was used as a lubricant has been investigated. First, the phenomenon of friction reduction was shown and different loads, speeds, and viscosities of lubricants were selected. Then, the lubrication state of this process was calculated and investigated, based on which the theoretical mechanism of the photo-modulation of friction lubrication was proposed. This study combines photosensitivity, photocatalysis, and friction and proposes a method to reduce the friction coefficient without changing the friction pairs or lubricants. A complete theoretical speculation is presented, revealing the role of photocatalysis in the lubrication process, which provides a new idea for the development of light-controlled friction.

## 2. Materials and Methods

The upper friction pair in the tribological experiment is a silicon nitride (Si_3_N_4_) ball with a diameter of 12 mm, purchased from Beijing Sinoma Synthetic Crystals Co., Ltd. (Beijing, China). The lower friction pair is a TiO_2_ plate with a diameter of 25.4 mm and a height of 5 mm, purchased from ZhongNuo Advanced Material (Beijing) Technology Co., Ltd. (Beijing, China). Table 1 shows the physical properties of these two materials. The sulfuric acid (H_2_SO_4_), petroleum ether, acetone, and anhydrous ethanol used in the experiments were purchased from Shanghai Aladdin Biochemical Technology Co., Ltd. (Shanghai, China). The PAO base oil, including PAO2, PAO4, PAO6, PAO8, and PAO10, was purchased from Shanghai Qicheng Lubricating Material Co., Ltd. (Shanghai, China). Table 2 shows the different viscosity values for the PAO.

The friction tests were carried out on a universal micro-tribotester (UMT5, Bruker, Billerica, MA, USA) in a rotational mode with a ball-on-disk configuration. Figure 1a shows the Schematic diagram of the experiment. Before each tribological experiment, Si_3_N_4_ balls and TiO_2_ plates were cleaned with acetone, anhydrous ethanol, and deionized water in turn via sonication for 10 min to remove contaminants from their surfaces.

All experiments were performed at room temperature of 25 °C and humidity of 30%. The solution for running-in was a sulfuric acid solution with a pH of 1. In total, 20 μL sulfuric acid solution was added before running-in and the running-in time was 5 min. The UV group of TiO_2_ was cleaned and then irradiated using a mercury lamp for 2 h after the running-in. The mercury lamp we used mainly emits at a wavelength of 400 nm and has an optical power of 80 mW/cm^2^. Afterward, 30 μL PAO base oil was added to the TiO_2_ surface. We used different base oils, PAO2, PAO4, PAO6, PAO8, PAO10, and the corresponding viscosities are shown in Table 2. In the experiment, the radius of rotation was 3 mm, and for the speed-changing experiment, the load was 3 N, the rotational speeds were 20–1200 rpm, and the corresponding linear speeds were 6.28–377.0 mm/s. The speed was set from high speed to low speed, and then from low speed to high speed again to ensure that there was no difference between them. For the test of load variation, the loads were 1–8 N and the speeds were 56.5 mm/s.

After the tribological experiment, the TiO_2_ plates and Si_3_N_4_ balls were rinsed with petroleum ether, ethanol, and deionized water and blown dry with an earwash ball. The topography and surface roughness of the ball and plate were investigated using the non-contact optical three-dimensional interference profilometer (Nexview, ZYGO Lamda, Middlefield, CT, USA). An X-ray photoelectron spectroscope (XPS, PHI Quantera II, Kanagawa, Japan) was used to characterize the elements in the wear surfaces.

Microscopic friction experiments were performed on wear zones of TiO_2_ plates using an atomic force microscope (AFM, Bruker Dimension Icon, Billerica, MA, USA). The probe was a self-made silica sphere probe, where a 22 μm-diameter silica sphere was glued on a Tipless silicon tip rectangular cantilever beam (TL-CONT, purchased from Nanosensors Co. Neuchatel, Switzerland). The normal elasticity coefficient of the probe was 0.2535 N/m and the lateral sensitivity coefficient of the laser was 199.3 nm/V. Each group of samples was tested 3–5 times, and each group of voltage data was averaged according to the stable value selected from the test results.

A quartz crystal microbalance (QCM, Q-Sense E4, Gothenburg, Sweden) was used to test the interfacial adsorption of PAO2 base oil molecules on the chip of a TiO_2_-coated quartz crystal. The instrumentation and chip needed to be cleaned repeatedly before the experiment using petroleum ether, alcohol, and ultra-pure water and dried by nitrogen blowing.

## 3. Results and Discussion

### 3.1. Friction Reduction after UV Treatment

A macroscopic tribology experiment was conducted on UMT5 and demonstrated the friction reduction achieved by UV treatment. The friction pairs chosen were the Si_3_N_4_ ball and TiO_2_ plate. Si_3_N_4_, with its high hardness, excellent wear resistance, corrosion resistance, and oxidation resistance, is an excellent material as a friction pair and is well suited for the study of the tribological properties of TiO_2_. As shown in Figure 1b, the friction coefficient stabilized at about 0.035 after 300 s of running the sulfuric acid solution on the TiO_2_ surface. After washing off the acid solution and blowing it dry, 30 μL of PAO4 base oil was added, and the experiment was conducted with a load of 3 N and a speed of 56.5 mm/s. The coefficient of friction stabilized at 0.034 and could be maintained for a long time (at least 4 h). If the TiO_2_ surface was illuminated by UV, the coefficient of friction stabilized at 0.016 and also maintained for a long time. The friction coefficient of the PAO4 oil-lubricated Si_3_N_4_-TiO_2_ friction system was reduced by 52.9% when treated with UV.

As shown in Figure 2a, when the experimental speed was set to 56.5 mm/s and the load was changed, on the TiO_2_ without UV treatment, the friction coefficient first decreased and then increased and reached the lowest when the load was 3 N. In the UV-treated group, the friction coefficient tended to decrease compared with the one without UV treatment and reached the lowest at 2 N. In Figure 2b, the load was set to 3 N and the speed during the experiment was changed. On the TiO_2_ without UV treatment, the overall friction coefficient was in the range of 0.01–0.015 as the speed became greater than 130 mm/s, while in the UV-treated group, the friction coefficient tended to be lower and even reached a superlubricity state (0.0062 at a speed of 377.0 mm/s). At speeds less than 130 mm/s, although the friction system did not show a transition from non-superlubricity to superlubricity after the UV treatment, a reducing trend could still be observed. At lower speeds of 22 mm/s and 31 mm/s, the coefficient of friction decreased from 0.0545 and 0.0473 to 0.0226 and 0.0196, respectively, with a reduction of 58.6% in both cases.

Since the viscosity of the lubricant is relatively high, it is easy to form a fluid lubrication film during the friction process and reduce the friction coefficient between systems, and for different friction systems, different viscosity lubricants can be used for lubrication. Therefore, it is important to investigate the effect of UV treatment on the regulation of PAO base oil with different viscosities.

Figure 2c–f shows the change in friction coefficient after UV treatment for four PAO base oils with different viscosities when lubricating the Si_3_N_4_-TiO_2_ friction system (all were first run with an acid solution on the friction pairs). The results showed that there was an obvious reduction of friction coefficient after UV treatment in PAO base oil solutions of different viscosities. When PAO2 base oil (10.1 mPa·s) was used, at higher speeds (>150 mm/s), the system went from non-superlubricity to superlubricity after UV treatment and achieved a lower friction coefficient (0.004). But, at lower speed intervals (<70 mm/s) the friction coefficients of the UV-treated and non-UV-treated groups were similar and both were at a high level (0.07–0.08). When lubricated with PAO6 base oil (49.9 mPa·s), in the speed range of 30–250 mm/s, the friction system could achieve the transition from non-superlubricity to superlubricity. The reduction was also obvious at higher speeds, with a range of 30%, and at lower speeds, the friction coefficient reduced more, which was reduced by 61.4% from 0.0335 to 0.0129 at a speed of 22 mm/s. The coefficient of friction was also reduced after UV treatment when lubricated by PAO8 base oil (87.5 mPa·s) and PAO10 base oil (108.6 mPa·s), but the percentage of reduction was relatively low. The friction coefficient on TiO_2_ with and without UV treatment both remained in the range of 0.01–0.02 in the tested speed interval.

### 3.2. Lubrication States When Reducing Friction

The classical Stribeck curve points out that as the bearing characteristic number increases, the friction system transitions from boundary lubrication to mixed lubrication and finally enters the stage of elastohydrodynamic lubrication, and the friction coefficient shows a trend of first decreasing and then slowly increasing. Since the factors and mechanisms affecting the friction system vary under different lubrication states, it is important to determine the lubrication state when lubricated with different base oils to reveal the mechanism of friction coefficient reduction after UV treatment. As shown in Figure 3a, the film thicknesses of different PAO base oils in the Si_3_N_4_-TiO_2_ friction system at different speeds were calculated according to Hamrock–Dowson (H-D) equations [37] as follows:(1)$$H_{c}^{*}=2.69\frac{G^{*0.53}u^{*0.67}}{W^{*0.067}}\left(1-0.61e^{-0.73k}\right)$$
where $H_{c}^{*}=h_{c}/R$, $G^{*}=\alpha E^{\prime }$, $\mu^{*}=\eta_{0}u/E^{\prime }R$, and $W^{*}=W/E^{\prime }R^{2}$. *h_c_* is the film thickness, *α* is the pressure–viscosity coefficient of the oil, *u* is the average linear speed of the TiO_2_ plate and Si_3_N_4_ ball, *W* is the load, *k* is a coefficient, and *E*′ is the reduced Young’s modulus of the two contacting solids,
(2)$$\frac{1}{E^{\prime }}=\frac{1}{2}\left(\frac{1-\mu_{1}^{2}}{E_{1}}+\frac{1-\mu_{2}^{2}}{E_{2}}\right)$$
where *E*_1_ and *E*_2_ are Young’s modulus of Si_3_N_4_ and TiO_2_, respectively. *μ*_1_ and *μ*_2_ are Poisson’s ratios of Si_3_N_4_ and TiO_2_, respectively, and *R* is the equivalent radius of the ball that could be described by the Hertz contact theory [38]:(3)$$R=\frac{E^{\prime }D^{3}}{6W}$$
where *D* is the diameter of the worn scar of the ball. The percentage reduction of friction coefficient after UV treatment at different speeds when lubricated with different PAO base oils was calculated and is shown in Figure 3b–f.

The results show that the reduction rate of the friction coefficient after UV treatment is closely related to the lubricant film thickness. For different PAO base oils, the median of the friction coefficient reduction was selected, and the film thickness of the lubricant when the reduction rate of the friction coefficient is larger than the median was calculated. As shown in Table 3, the median reduction rate of the friction coefficient after UV treatment for different PAO base oil lubricants and the interval of film thickness corresponding to the reduction rate larger than the median are calculated, respectively.

The results show that the high reduction in the friction coefficient mainly corresponds to the same film thickness interval, basically between 30 nm and 100 nm, despite the different viscosities and speeds. The lubrication state could be distinguished by the ratio of the theoretical film thickness to the equivalent surface roughness (Rq), and the ratio *λ* could be calculated by the following formula:(4)$$\lambda =\frac{h_{c}}{\sqrt{\sigma_{1}^{2}+\sigma_{2}^{2}}}$$
where *σ*_1_ and *σ*_2_ are the surface roughness, i.e., the root mean square average of the profile heights over the evaluation length (Rq) of the worn regions of the Si_3_N_4_ balls and TiO_2_ plates, respectively. The system was in the elastohydrodynamic lubrication state if the ratio *λ* exceeded three, mixed lubrication state if the ratio *λ* was between one and three, and boundary lubrication state if the ratio *λ* was less than one. According to the characterization of the upper and lower friction surfaces after the experiment, it was found that the Rq of the upper friction Si_3_N_4_ was 3.5 nm, the Rq of the lower friction TiO_2_ plate was 29.5 nm, and the equivalent roughness *σ** was 29.7 nm; thus, the ratio λ value was 1.01 to 3.37, which was consistent with the *λ* value range 1–3 corresponding to the mixed lubrication condition. This shows that in the PAO base oil lubrication with different viscosities, the friction coefficient decreases most significantly when the system is in a mixed lubrication condition.

When the system is in a mixed lubrication state, UV treatment shows the most obvious effect. The mixed lubrication state includes elastohydrodynamic lubrication and boundary lubrication, and according to the research conducted by Han et al. [35], when using water-based lubricants, such as glycerol solution, UV light modulates the friction and lubrication of the system mainly by interfacial adsorption of molecular films; in other words, the part of boundary lubrication is regulated, while elastohydrodynamic lubrication is not much influenced. However, compared to water-based lubricants, oil-based lubricants have higher viscous pressure coefficients (18 GPa for PAO4 and 6 GPa for propylene glycol), i.e., the state of the lubricant is more influenced by the experimental conditions during oil lubrication. In Figure 2a, the load affects the friction coefficient significantly greater than UV light does. Therefore, it is necessary to discuss the effect of the elastohydrodynamic lubrication part in the study of UV light on the lubrication of PAO base oil.

Since the surface of Si_3_N_4_-TiO_2_ friction pairs were treated with an aqueous solution of acid before oil lubrication, the actual contact area was much larger than the calculated value of the point–surface contact model, so the contact pressure of the system could no longer be calculated by using the Hertz contact model. As shown in Figure 4, the diameter of wear scars on the Si_3_N_4_ ball surface were therefore used to calculate the actual contact pressure. After the experiments, the diameters of the wear scars were 578.5 μm without UV treatment and 525.2 μm with UV treatment, which corresponded to a calculated actual contact pressure of 11.4 MPa and 13.8 MPa, respectively.

In the case of fluid lubrication, the friction coefficient *μ* of the system can be calculated using the following equation
(5)$$\mu =\frac{\tau A}{W}$$
where *A* is the contact area between the upper and lower friction pairs during lubrication, *W* is the load in the experiment (3 N), *τ* is the shear pressure of the lubricating fluid between the friction pairs, and *A* and *τ* can be calculated by the following equation:(6)$$\tau =\eta_{0}e^{\alpha p\frac{u}{h_{c}}}$$
(7)$$A=\pi \left(\frac{3RW}{4E^{*}}\right)$$
(8)$$p=\frac{W}{A}$$
where *η*_0_ is the viscosity of the lubricant, *α* is the pressure–viscosity coefficient of the lubricant, *p* is the average contact pressure, *u* is the average linear velocity (i.e., the velocity in the experiment, which is taken as 56.5 mm/s in this calculation), *h_c_* is the calculated film thickness, *R* is the equivalent contact radius, and *E** is the equivalent elastic modulus of the two friction pairs.

According to the above equation, the coefficient of friction μ_1_ is calculated as 0.0047 for the experiment without UV treatment and 0.0037 for that with UV treatment, while the actual friction coefficients are 0.034 and 0.016. The difference between the two is mainly because it is considered to be an overall elastohydrodynamic lubrication in the calculating process, while the actual experimental conditions had mixed lubrication in the system and there was a contribution of both elastohydrodynamic and boundary lubrication. However, it can be inferred from the calculation of the friction coefficient that from the point of view of the viscous pressure coefficient, the difference in pressure values observed in this experiment is not sufficient to bring about a significant reduction in the friction coefficient.

On the other hand, the study by Jinjin Li et al. [39] pointed out that the running-in process not only makes the overall contact pressure of the system lower but also causes the upper and lower friction pairs to form a non-parallel surface, as shown in Figure 5a. There exists a certain inclination angle (*θ*) for the upper friction pair, and this inclination angle can help the system to form an elastohydrodynamic film during the elastohydrodynamic lubrication process, thus realizing the elastohydrodynamic effect. As shown in Figure 5b,c, the wear scars on the Si_3_N_4_ balls were measured. It was observed that there was indeed a slope with a certain angle, as shown in Figure 5d,e, and the corresponding inclination angles were 0.017° and 0.024° for θ_1_ and θ_2,_ respectively.

The shear force and friction coefficient can be calculated using the Reynolds equation as follows [40]:(9)$$\frac{\partial }{\partial x}\left(h_{c}^{3}\frac{\partial p}{\partial x}\right)+\frac{\partial }{\partial y}\left(h_{c}^{3}\frac{\partial p}{\partial y}\right)=6u\eta \frac{dh_{c}}{dx}$$
where *x* denotes the velocity direction, *y* denotes the vertical paper direction (take Figure 5a as an example), and *η* is the viscosity of the lubricant. Since the role of the viscous pressure effect is negligible in this experiment, as discussed earlier, the value of *η*_0_ is taken. Solving this equation yields the friction coefficient *μ* [39]:(10)$$\mu =\left(\frac{C_{\mu }}{\sqrt{C_{w}}}\right)\sqrt{\frac{\eta_{0}uL}{W}}$$
(11)$$C_{w}=\frac{6}{{\left(m-1\right)}^{2}}\left[\ln m-\frac{2\left(m-1\right)}{m+1}\right]$$
(12)$$C_{\mu }=\frac{4\ln m}{m-1}-\frac{6}{m+1}$$
where *L* denotes the length of the contact surface in the x-direction and can be considered as the diameter of the wear scar *D*. *m* is the ratio of the film thickness at the entry *h*_1_ to the film thickness at the exit *h*_2_.

On this basis, it is calculated that the friction coefficient *μ*_1_ is 0.0012 for the friction pairs without UV treatment and *μ*_2_ is 0.0010 for that after UV treatment, which are also far from the actual friction coefficient values of 0.034 and 0.016. Therefore, the results of this calculation indicate that the slope angle formed during the running-in process cannot explain the significant decrease in the friction coefficient after UV treatment.

From the above analysis, it can be found that the elastohydrodynamic lubrication part of the mixed lubrication has less effect on the system friction coefficient, so the mechanism of UV light on friction lubrication must be further analyzed from the boundary lubrication part.

### 3.3. Mechanism of Friction Reduction: Photocatalysis

After the macroscopic experiments were completed, the friction coefficient of the wear scar on the surface of the TiO_2_ was tested in a dry environment using an AFM and a silica ball probe. The Schematic diagram of the test is shown in Figure 6a. After the UMT5 friction experiment using PAO4 as a lubricant, the TiO_2_ surface was washed with petroleum ether, alcohol, and deionized water, and then the surface was blown dry and finally placed on the AFM bench for microscopic friction experiments. Figure 6b demonstrates several results of the AFM tribology tests, where the gray and pink areas show the results of TiO_2_ without and with UV treatment. The friction coefficient (slope of the data) was 0.223 and 0.405 for experiments, respectively, and this result is consistent with the trend of the macroscopic experiments in Figure 1. Figure 6c,d shows that the microstructures of the surface with and without UV treatment are very similar. Since the microscopic experiments were conducted in an air environment, i.e., the friction process involved direct contact between the silica spheres and the wear scar of the TiO_2_ plate, the cause of the difference in friction force can only come from the surface of the TiO_2_ wear scar. The only possible source of lubrication is the PAO4 oil during the UMT5 macroscopic friction experiment. Therefore, it is presumed that the phenomenon of the reduction in the friction coefficient after UV treatment in both macroscopic and microscopic experiments during oil lubrication comes from the enhanced adsorption of PAO base oil molecules on the surface of inorganic photosensitive materials occurring after UV treatment, forming a lubricating film with a good lubrication effect.

It is well known that PAO base oils mainly consist of hydrocarbons. However, as shown in Figure 7a, the XPS spectra found that without UV treatment the C signal was dominated by C–C (284.8 eV) with 96.2% of the signal, and about 3.8% of the signal was from C–O (286.1 eV). However, this proportion changed considerably after UV treatment. As shown in Figure 7b, although the C signal in C–C (284.5 eV) was still the most abundant, occupying 69.5%, the proportion of the C signal in C–O (285.7 eV) increased significantly to 19.3%, in addition to 11.2% of the C signal from C=O. The statistics were deconvoluted using the software OriginPro 2018C, via the Gaussian method.

The results of XPS demonstrated that the residual PAO molecules on the TiO_2_ surface were no longer purely C–H and C–C structures, but rather C–O structures or even C=O bonds. The appearance of these structures is attributed to another important application of inorganic photosensitive materials: photocatalysis. When the light energy absorbed by inorganic photosensitive materials (such as TiO_2_, ZnO, and WO_3_) is greater than their threshold band gap energy, the electrons on their valence bands are excited, generating a large number of electron–hole pairs which are transferred to the surface of the materials and undergo redox reactions with the substances on the surface, playing a photocatalytic role, and this property is often utilized for the degradation of organic substances such as hydrocarbons and phenols [20,21,41]. In the case of photocatalytic degradation of methane on the TiO_2_ surface, for example, under the action of light field, electron–hole pairs with high activity are transferred to the TiO_2_ surface. Photogenerated electrons reduce oxygen to produce ·O_2_^−^ or ·OOH and holes oxidize H_2_O and CH_4_ to produce ·OH and ·CH_3_, and eventually these radicals further generate CH_3_OOH and CH_3_OH [24,25,42].

In this experiment, the XPS spectra of Figure 7 show that the PAO base oil molecules on the TiO_2_ surface produce C–O and C=O structures internally, and this provides the possibility for the adsorption of PAO base oil molecules on the TiO_2_ surface. It forms a hydrogen bond structure with Ti–OH on the TiO_2_ surface which then adsorbs, making the friction force in the boundary lubrication part of the mixed lubrication much less than the direct shear resistance between the rough peaks, thus making the friction coefficient of the system significantly lower after the UV treatment.

To prove this conjecture, Figure 8 tests the adsorption of PAO base oil molecules on the SiO_2_ surface and the TiO_2_ surface before and after UV treatment using a QCM. Since the viscosity of the liquid has a large effect on the value of the frequency change during the test, a PAO2 base oil with a lower viscosity was chosen for the experiments. A solution of PAO2 with a mass fraction of 20% was prepared using petroleum ether (viscosity of 0.981 mPa·s) as the solvent, with a viscosity value of 2.08 mPa·s.

In the experiment, petroleum ether, petroleum ether solution of PAO2, and petroleum ether flowed sequentially onto the chip of a TiO_2_-coated quartz crystal. Triggered by an alternating current, the vibration of the chip and its attenuation was recorded and the resonance frequency was obtained. When different liquid flows over the chip, the substances adsorbed to the surface may affect the frequency [43,44]. In Figure 8a, ∆f_1_ in the purple stage represented the total adsorption of PAO2 base oil molecules on the chip surface, and here, ∆f_1_ was −2302.9 Hz. On the TiO_2_ surface without UV treatment, this value became −2645.6 Hz, while on the UV-treated TiO_2_, this value became −4650.6 Hz, shown in ∆f_1_ and ∆f_2_ in Figure 8b. In the case of the same test liquid, the difference due to the liquid itself was negligible. Then, the effect of UV treatment on the frequency change in the TiO_2_ chip was caused by the change in adsorption of PAO2 molecules. Further evidence was provided in the second gray stage in Figure 8a: ∆f_2_ on the SiO_2_ surface represented the interfacial adsorption of PAO2 base oil molecules on the SiO_2_ chip, which was greater than the attraction of the petroleum ether solvent and therefore changed the measured frequency. This value reached −321.1 Hz on the SiO_2_ surface, which was much smaller in absolute value than Δf_1_, proving that the interfacial adsorption of PAO2 base oil molecules accounted for a small fraction on the SiO_2_ surface, while on the TiO_2_ surface without UV treatment, the Δf_3_ value representing the interfacial adsorption was −959.8 Hz; this value was larger than that on the SiO_2_ surface, indicating stronger interfacial adsorption on the TiO_2_ surface. In the UV group, the corresponding value of Δf_4_ was −1885.6 Hz, which was much larger than Δf_3_, indicating stronger interfacial adsorption of PAO2 base oil molecules after UV treatment. This is consistent with the previous analysis that the adsorption of PAO base oil molecules on the UV-treated TiO_2_ surface was stronger than that on the non-UV-treated surface, which is the essence of the reduced friction coefficient of the system after UV treatment.

To further verify the effect of photocatalysis on the modulation of the friction coefficient of the system, a set of photocatalytic group experiments was added to the original experiments in this study. The experimental procedure was as follows: the same acid solution was run on Si_3_N_4_ and TiO_2_ friction pairs, and the surface was cleaned and blown dry after the running-in, followed by the addition of 30 μL of PAO4 base oil and then a 2 h UV treatment. Compared with the original experiment in Section 3.1, the order of UV treatment and the addition of oil was switched.

Figure 9a shows the experimental results on TiO_2_ with and without UV treatment and the photocatalytic group, where the load is 3 N. The results showed that the friction coefficients of the photocatalytic group and the UV-treated group were close to each other at speeds greater than 200 mm/s, and they both decreased compared to the non-UV-treated group. At speeds between 50 mm/s and 200 mm/s, the friction coefficients of the UV-treated group decreased significantly, and even the transition from non-superlubricity to superlubricity was achieved. This result is consistent with the previous analysis, i.e., the elastohydrodynamic part contributes little to the friction coefficient and the reduction of the friction coefficient in the mixed lubrication stage is greater. There were more hydroxyl, aldehyde, and carboxyl groups in the photocatalytic group; so, the adsorption ability on the surface is stronger and the reduction of the friction coefficient is more obvious.

Figure 9b shows the variation of the friction coefficient and the proportion of the C–O and C=O signal to the carbon XPS energy spectrum signal at the wear scar of the Si_3_N_4_-TiO_2_ friction system under PAO4 lubrication after different UV treatment times. The results showed that the friction coefficient gradually decreased with the increase in the proportion of C in the C–O and C=O to the overall C signal, which was consistent with the previous speculation on the mechanism. With the increase in UV treatment time, more PAO base oil molecules were photocatalyzed, more C–O and C=O bonds were formed, more molecules could be adsorbed on the TiO_2_ surface, and a better lubrication effect of the molecular adsorption film formed, which eventually led to the reduction in the friction coefficient.

After the above discussion and analysis, this study proposes the mechanism of friction reduction achieved by UV radiation on a TiO_2_ surface, as shown in Figure 10. In the area where the rough peaks of the Si_3_N_4_ sphere and TiO_2_ plate are in direct contact with each other when no UV treatment is performed, the effective oil film cannot be formed between the rough peaks due to the pressure and the poor adsorption ability of PAO molecules on the surface so the rough peaks directly crash and shear, which makes the friction coefficient of the system relatively high. After the UV treatment, due to the photosensitivity of TiO_2_, electron–hole pairs are generated on the surface and redox reactions occur with PAO molecules. Some C–O bonds in PAO molecules are replaced by hydroxyl, aldehyde, or carboxyl groups, which build hydrogen bonds or occupy oxygen vacancies with hydroxyl structures on the upper and lower friction pairs and are finally adsorbed on the surface. In the process of friction, the direct contact of rough peaks becomes the contact of rough peaks and the adsorbed molecular layer. This greatly reduces the friction force and finally reduces the friction coefficient of the system, achieving the purpose of regulating the friction coefficient by light field.

## 4. Conclusions

In this work, an obvious friction reduction has been achieved by UV illumination on inorganic photosensitive material TiO_2_. The modulating behavior and laws of modulation were investigated, and the related mechanisms were analyzed. It was found that when using PAO4 base oil as lubrication, the UV treatment was able to reduce the friction coefficient from 0.034 to 0.016, with a reduction of 52.9%, at a load of 3 N and a speed of 56.5 mm/s. Based on the comparison of the calculated values of surface roughness and film thickness, it was found that the best regulation effect was achieved when the system was in a mixed lubrication stage. The mechanism of light field modulation lies in the photocatalytic properties of TiO_2_. The oxidized PAO molecules adsorb to the TiO_2_ surface after UV treatment, forming an adsorption layer which avoids the direct contact of rough peaks and thus reduces the friction coefficient.

In this paper, a significant reduction in the friction coefficient greater than 50% was achieved using UV illumination, making friction regulation no longer dependent on changes in the friction pairs or lubricant. The friction reduction effect at different loads and speeds was investigated and the mechanism of photo-induced friction coefficient change is also proposed from the photocatalytic point of view, which provides a new idea for light-controlled friction. This study fills the gap of previous research and systematically investigates the relationship between TiO_2_ photocatalysis and lubrication. There are still some problems to be solved, such as the long time required for light treatment and the harsh run-in conditions. In the future, we will continue to work in this direction to improve the efficiency of light response and explore its applications in the engineering field.

## Figures and Tables

**Figure 1 materials-17-01680-f001:**
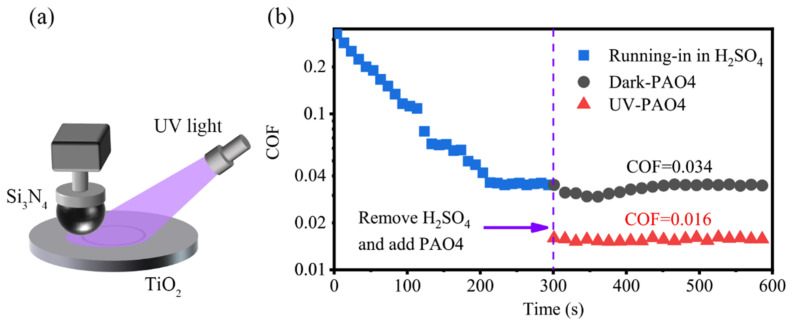
(**a**) Schematic diagram of UMT experiment; (**b**) variation of friction coefficient of TiO_2_ surface when PAO4 base oil was used as lubricant after acid running-in with and without UV treatment.

**Figure 2 materials-17-01680-f002:**
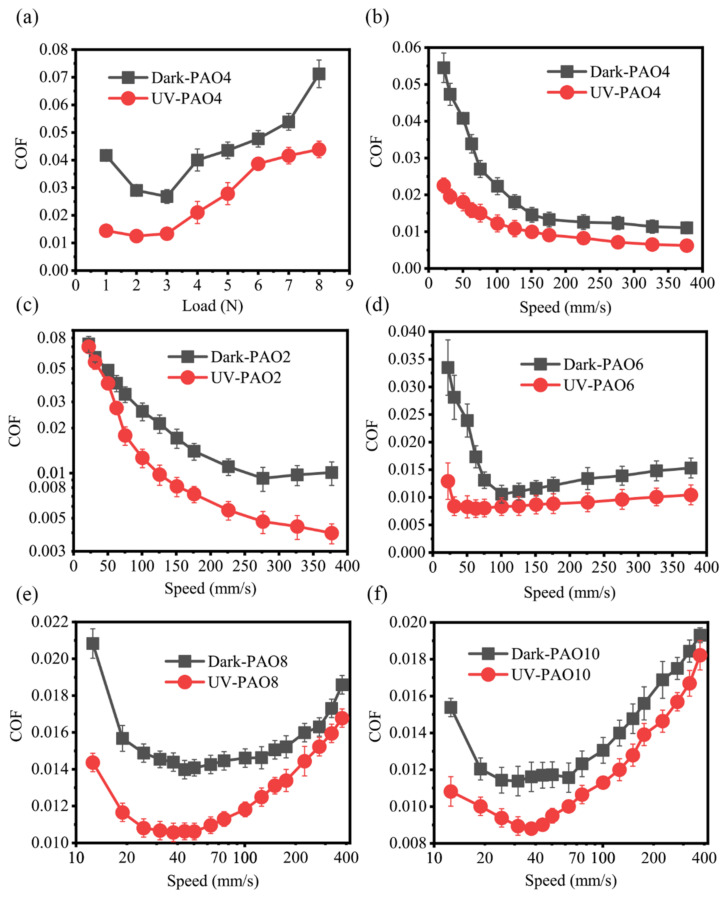
Variation of friction coefficient (**a**) with load (**b**) and speed of TiO_2_ surface when PAO4 base oil was used as lubricant after acid running-in; variation of friction coefficient with speed of TiO_2_ surface after UV treatment when lubricated by different PAO base oils after acid running-in: (**c**) PAO2; (**d**) PAO6; (**e**) PAO8; (**f**) PAO10.

**Figure 3 materials-17-01680-f003:**
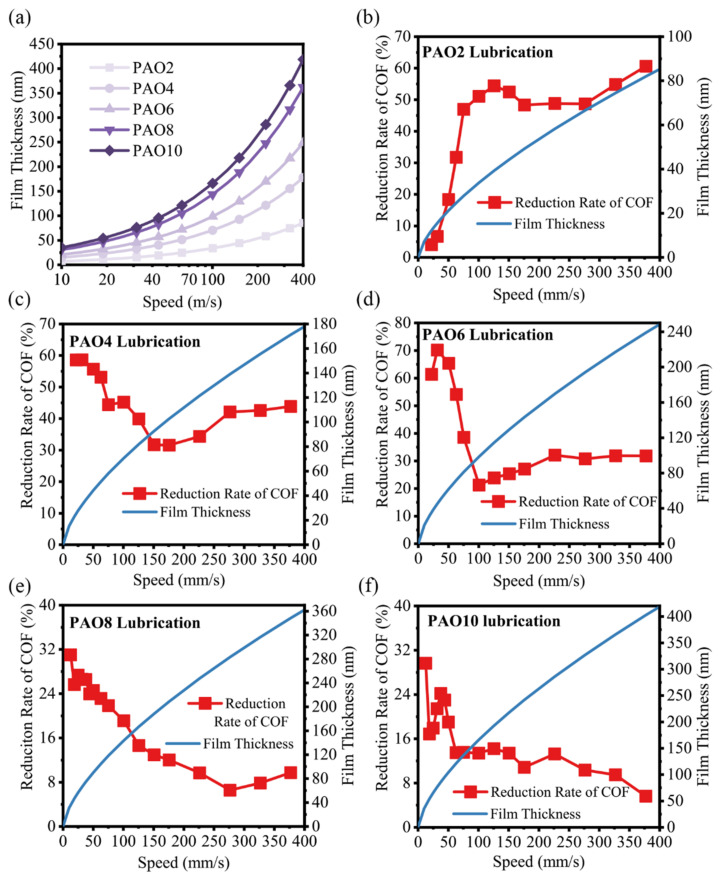
(**a**) Calculated values of film thickness at different speeds for PAO lubricants with different viscosities; correspondence between the reduction ratio of friction coefficient and calculated values of lubricant film thickness at different speeds for (**b**) PAO2; (**c**) PAO4; (**d**) PAO6; (**e**) PAO8; (**f**) PAO10 base oil lubrication.

**Figure 4 materials-17-01680-f004:**
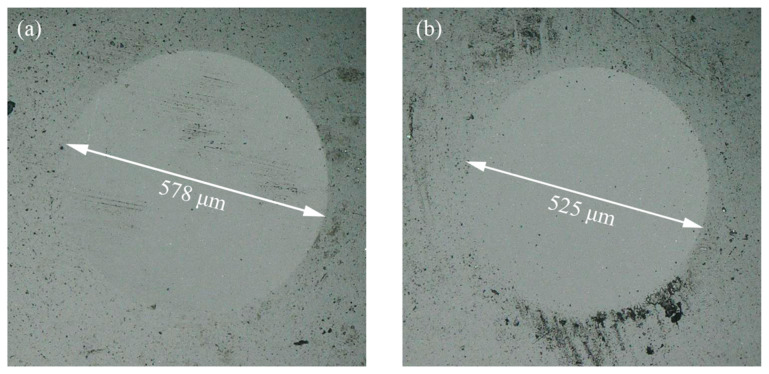
Optical pictures of Si_3_N_4_ spherical wear spots after the experiment (**a**) without UV treatment and (**b**) with UV treatment.

**Figure 5 materials-17-01680-f005:**
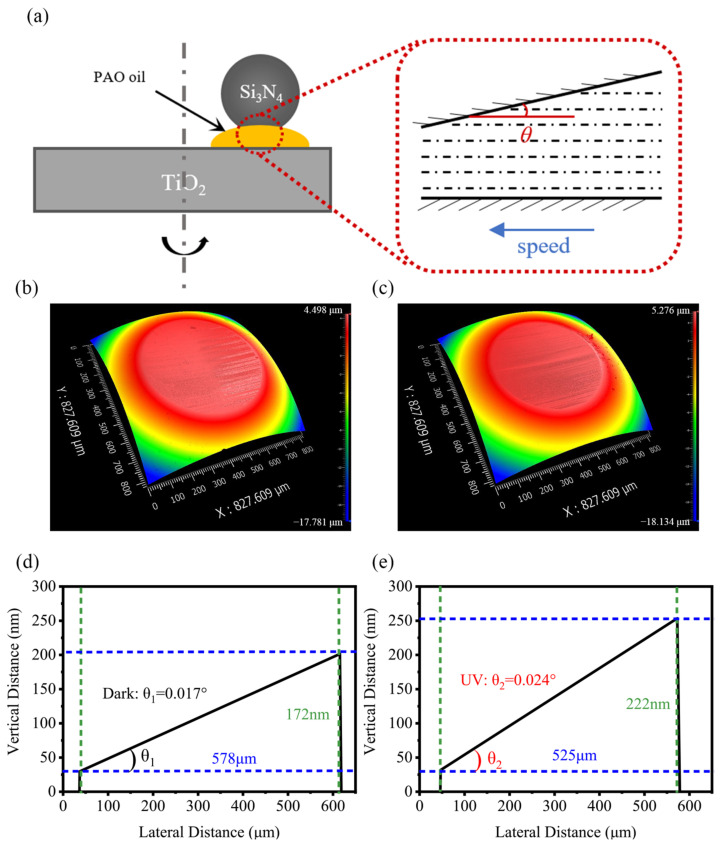
(**a**) Schematic diagram of lubrication model. Wear scars on Si_3_N_4_ balls (**b**) without UV treatment and (**c**) with UV treatment. Slope generated by Si_3_N_4_ ball on the wear scar along the velocity direction (**d**) without UV treatment and (**e**) with UV treatment.

**Figure 6 materials-17-01680-f006:**
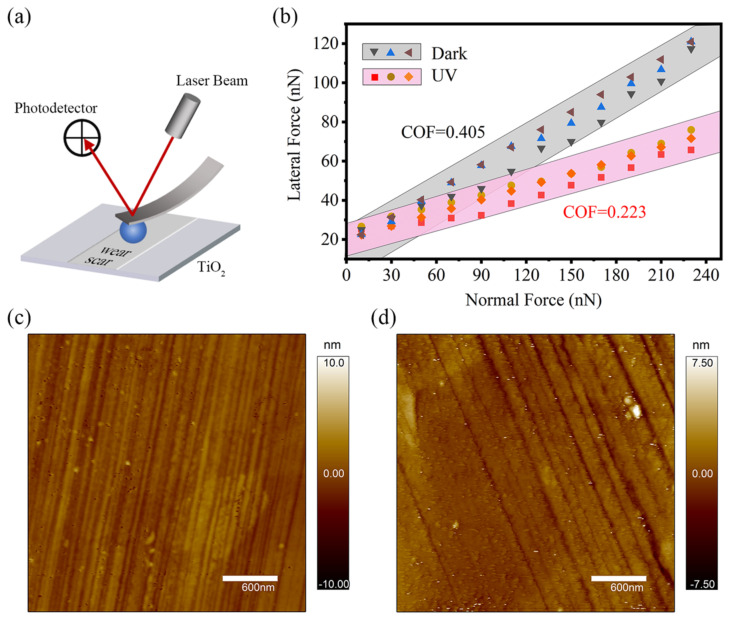
(**a**) Schematic diagram of the AFM test; (**b**) microscopic friction coefficient values of TiO_2_ wear scar after experiments measured using AFM. Microstructure of the wear zone (**c**) with and (**d**) without UV treatment.

**Figure 7 materials-17-01680-f007:**
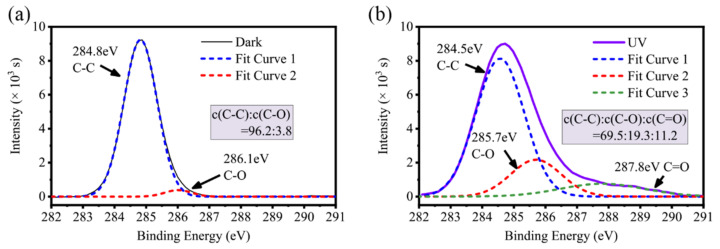
XPS spectra of C in the wear scar of TiO_2_ plates after the experiment (**a**) without UV treatment and (**b**) with UV treatment.

**Figure 8 materials-17-01680-f008:**
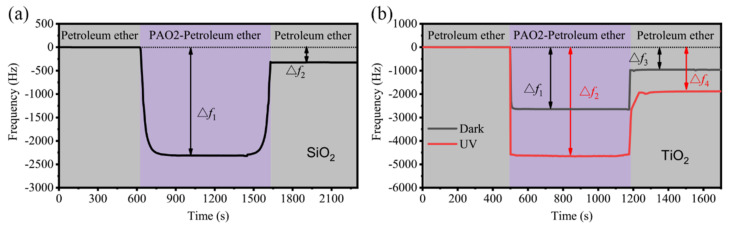
Adsorption of PAO2 mixed with petroleum ether on (**a**) SiO_2_ surface and (**b**) TiO_2_ surface with and without UV treatment.

**Figure 9 materials-17-01680-f009:**
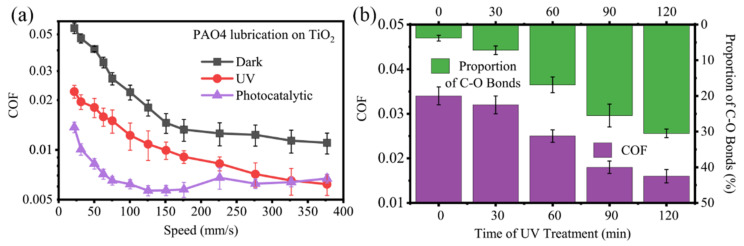
(**a**) Variation of friction coefficient with velocity on TiO_2_ with and without UV treatment and photocatalytic group on the running-in samples with PAO4 base oil lubrication; (**b**) variation of friction coefficient and the proportion of carbon–oxygen bond signal to carbon XPS energy spectrum signal at the wear scar of Si_3_N_4_-TiO_2_ friction system under PAO4 lubrication with different UV treatment times.

**Figure 10 materials-17-01680-f010:**
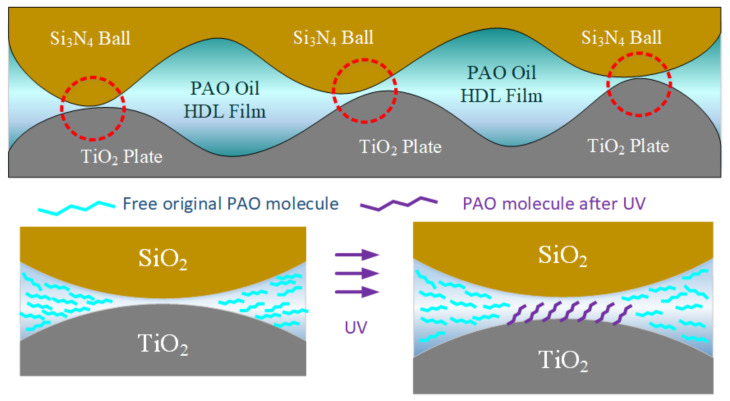
Schematic diagram of the lubrication mechanism of photomodulated PAO base oil. Red circles indicate the contact area.

**Table 1 materials-17-01680-t001:** Physical properties of the friction pairs [36].

Material	Si_3_N_4_	TiO_2_
Young’s modulus (GPa)	310	230
Poisson’s ration	0.26	0.17

**Table 2 materials-17-01680-t002:** Viscosity values of different PAO reagents used in the experiments.

Polyalphaolefin Type	PAO2	PAO4	PAO6	PAO8	PAO10
Viscosity (mPa·s)	10.1	30.3	49.9	87.5	108.6

**Table 3 materials-17-01680-t003:** Median reduction rate of friction coefficient after UV treatment of different lubricants and the corresponding film thickness when larger than the median.

PAO Base Oil Types	Median Reduction Rate of Friction Coefficient	Corresponding Film Thickness Interval (nm)
PAO2	48.30%	33.8–82.0
PAO4	43.80%	27.7–70.6
PAO6	31.90%	38.9–81.3
PAO8	21.80%	35.6–104.8
PAO10	14.20%	41.2–104.4

## Data Availability

The data in this study are presented in the article.

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
