# Peer review of "Friction Reduction Achieved by Ultraviolet Illumination on TiO2 Surface"

_materials, 2024, doi:10.3390/ma17071680_

Round 1

Reviewer 1 Report

Comments and Suggestions for Authors

In this manuscript entitled "  Friction reduction achieved by ultraviolet illumination on TiO2 surface", authors mentioned about Friction reduction which was achieved by ultraviolet illumination on TiO2  surface. There are some improvements can be done before it can be accepted for publication.

In abstract, need to add some numerical orientated results too.

In keywords, need to add some catchy words

Introduction need to revise with some relevant references with adding few new paragraph in which author have to discuss in more details in dynamic way.

There is lack of scientific way of writing in whole section such as heading and sub-headings are even very general. so need to present the whole manuscript in a scientific way.

Need to add about the novelty of the work chosen and reported in this article.

Need to add some future prospects.

Please take about the typo, grammatical errors throughout the article.

Comments on the Quality of English Language

Need to improve the quality of English language.

Reviewer 2 Report

Comments and Suggestions for Authors

This study examines the effect of ultraviolet irradiation on the friction behaviour of titanium dioxide (TiO2) in sliding contact with silicon nitride (Si3N4). The authors compared the friction of UV-treated TiO2 with untreated TiO2 in the presence of various PAO base oil lubricants. The UV treatment appears to reduce the steady state friction coefficients, which was attributed to enhanced adsorption of PAO molecules following UV illumination.

Although this is an interesting study, I have some suggested revisions, details of which are given below.

1. The authors used Si3N4 balls on TiO2 discs to study the friction behaviour. However, they do not discuss the rationale for choosing this combination. What potential application does this seek to replicate? Why were the test conditions chosen? Was it to simulate a particular tribological environment?

2. In a previous paper co-written by two of the present authors (Han et al., Front. Mater., 10 (2023), 1109890), it was stated that the running-in procedure was necessary in order to achieve superlubricity conditions. As a result, in the present study they employed a running-in procedure using a sulphuric acid solution. Although this seems to be effective in the present study, how practical would that be for a real application?

3. Section 2: are there more details of the UV treatment that the authors could provide? All it says at present is that the specimens were “…cleaned and then treated by UV illumination for 2 hours…”.

3. In Section 3.2, where the authors discuss the film thickness calculations (pages 5/6), a table listing the relevant material properties of Si3N4 and TiO2 would be useful.

4. In various places in the text, the authors discuss surface roughness. However, they do not quote which surface roughness parameter. Is it Ra? Rq? Or something else? In Equation 4, the roughness parameters should be Rq.  The authors need to be more specific when discussing surface roughness.

5. Figure 6(b): Need to explain the symbols used in the graph.

5. There are some references (not cited) that should be included. They are: B.L. Perotti et al., “Phototribology: control of friction by light”, ACS Appl. Mater. Interfaces, 13 (2021), 43746-43654; and M. Sakai et al., “Reduction of fluid friction on the surface coated with TiO2 photocatalyst under UV illumination”, J. Mater. Sci., 47 (2012), 8167-8173.

7. Figure 7: What fitting procedure was employed to deconvolute the peaks in this figure?

8. Some of the Discussion section seems to be lacking in references: for example, many of the equations, as well as the discussion of the role of photocatalysis (lines 323-336). It would be beneficial for any reader seeking to understand this subject better if the authors could provide suitable references to support their arguments. This is equally true in other places of the manuscript.

9. The authors need to provide more explanation of the QCM measurements (p. 12). The frequencies are not properly introduced or adequately explained, neither is the test itself where the authors describe various solutions flowing “into the pipe”. The authors need to provide a short description of the measurement and what it tells us.

10. In the description of the “photocatalytic group experiments” in the last paragraph of page 12, it is not clear what the difference is between the photocatalytic group and the UV treated specimens.

Comments on the Quality of English Language

1. Introduction (line 54): please change to “…TiO2 was selected…”

2. Section 3.1 (line 109): please change to “…UMT5 demonstrated…”

3. Section 3.2 (line 164): Stribeck should begin with a capital “S”.

4. Section 3.2 (line 223): please change to “…the load affects…”

5. Section 3.3 (line 323): please change to “…XPS demonstrated…”

6. Section 3.3 (line 342): it is not clear what the authors mean by their phrase “rough peak solids”. Do they mean the interacting asperities of the two sliding surfaces?

7. Section 3.3 (line 371): please change to “This is consistent with the previous analysis…”

Reviewer 3 Report

Comments and Suggestions for Authors

Ref_comments to the paper titled as “Friction reduction achieved by ultraviolet illumination on TiO2 surface” written by the authors: Xiao Sang, Ke Han, Manfu Zhu and Liran Ma.

It is well known that the TiO2 materials are studied by the different technical and scientific teams. But, the question connected with the UV treatment of this type of the materials is interesting and has some unique practical interest. From this point of view the current article is actual and modern.

For the first, the authors have made the literature search connected with the analysis of 29 references. The manuscripts published last 5 years have been included and analyzed in this consideration as well. Thus, the author is known the problem and the author is a specialist in the TiO2 properties changing via UV treatment.

The paper is good illustrated via graphic and model data. The instrumentations are possible to find and established the important results as well. The discussion used in this manuscript is coincided with our basic physic-chemical knowledge.

Some recommendations (questions) are the followings:

1). Please add in your manuscript the data on transmittance and reflection change after the UV treatment. In this concern, please explain in detail what kind of the UV treatment has been used in your experiments? Please indicate the wavelength: Is it 126 nm, 172 nm, what else spectral range?

2). In Figure 6 (c and d) the authors have obtained the grating after the UV treatment of the studied materials. Please explain, how is the diffraction properties of your modified surfaces can be used in order to add an additional mechanism to attenuate the light?

3). Yes, it is true, that UV treatment can change the friction with good advantage. Please add the comparative results (table) from other scientific articles, when the author also can activate the UV treatment in order to decrease the friction.

4). I would like to ask the author, have you the data on the refractive properties changing after the UV treatment of the materials studied

Conclusion should be extended.

As for my local opinion, this paper can be published after the major revision.

Comments on the Quality of English Language

It is good.

Round 2

Reviewer 2 Report

Comments and Suggestions for Authors

The authors have made the changes I suggested. This paper can be accepted once they have made a small number of minor amendments, details of which are given below.

1. Page 2: Please give the references for the property values listed in Table 1.

2. Page 7: The table on this page should be Table 3 (see also the mention of the Table in line 202).

3. Figure 7: In their reply to my previous comments, the authors stated that they used Origin to deconvolute the XPD peaks. It would be helpful if they included this information in their discussion of the XPS results on page 11 (the paragraph starting on line 325).

4. In the Introduction, the authors mention that TiO2 has been used as an additive for lubricating oils to enhance their tribological behaviour. Another similar study (not referenced) is that by Birleanu et al., Scientific Reports, 12 (2022), 5201. It would be worth adding this to the references in the discussion (lines 49-51).
